# The Current State of Knowledge about the Biological Activity of Different Parts of Capers

**DOI:** 10.3390/nu15030623

**Published:** 2023-01-25

**Authors:** Beata Olas

**Affiliations:** Department of General Biochemistry, Faculty of Biology and Environmental Protection, University of Lodz, Pomorska 141/3, 90-236 Lodz, Poland; beata.olas@biol.uni.lodz.pl; Tel./Fax: +48-42-6354485

**Keywords:** caper, phenolic compounds, safety

## Abstract

The caper, from the Latin *capra*, meaning *goat*, is the common name for the salt-fermented floral buds of the perennial shrubs of the *Capparis* genus (*Capparacea* family). This genus is represented by about 250 species, including the very popular *C. spinosa* L. While the whole plant is edible, the aromatic floral buds are most widely consumed, being collected by hand prior to blooming, dried in the sun and pickled. Capers are usually served marinated in vinegar, brine or oil. They have a significant potential as dietary supplements due to their low calorie content and richness in bioactive phytochemicals. Numerous in vitro and in vivo studies have demonstrated that *C. spinosa* have various nutritional and biological properties, including antioxidant activity resulting from the presence of phenolic compounds. The present paper reviews the current literature concerning the biological properties of the fruits, buds, seeds, roots and leaves of *C. spinosa*, including their toxicity.

## 1. Introduction

Capers, named after the Latin *capra*, meaning *goat*, are the salt-fermented floral buds picked from the perennial shrubs of the *Capparis* genus (*Cappariadacea* Juss. family) [1,2,3]. This genus comprises about 250 species, including the very popular *Capparis spinosa* L., *Capparis decidua* (Forssk.) Edgew (commonly known as karira), and *Capparis ovata* var. ovata, and the less known *Capparis tomentosa* Lam., and *Capparis sepiaria* L. *Capparis spinosa* is one of the most important economic species in the *Cappariadacea* family, and is even mentioned in the Bible (Book of Ecclesiastes) [1,2,3]. 

Capers are grown all over the world, especially in the Mediterranean basin and Asia Minor. The main producers are Italy, France, Spain, Turkey, Greece, and Morocco. Of these, Turkey produces about 4500 tons a year [1,2,3]. *C. spinosa* hence has many names, being known as Alcapparo in Spain, Alaf-e-Mar in Persian, Cappero in Italy [1,2,3] and wild watermelon in China [4].

*C. spinosa* has oval and petiolate leaves, and the flowers are set singly on long peduncles growing from the base of the petiole. The flowers are usually white or pink and up to 7 cm in diameter. The plant blooms in July and August. The fruit is a multi-seeded berry containing about 36% lipids and 18% proteins. While the whole plant is edible, the most extensively consumed part is the aromatic floral bud, collected by hand prior to blooming, then dried in the sun and pickled. Capers are classified on the basis of size, from peppercorn to olive size, with the smaller sizes being more aromatic and having higher value [1,2,3]. 

After picking, capers are usually marinated in vinegar, brine or oil, and then fermented to eliminate any undesirable sour or pungent flavors and increase shelf life. The buds have very intense flavors, mainly consisting of sulfur compounds such as benzyl and methyl isothiocyanate. In addition to their valuable nutritional properties, capers do not contain a large number of calories: 100 g of capers only provides 23 kcal [1,2,3].

In southern Europe, capers were consumed more than two thousand years ago. Pickled capers are consumed in large quantities in the Mediterranean regions, and can be found in various products. They are key ingredients in a number of dishes, including poultry, fish, pizza or pasta. For example, in the United States, they are often added as flavoring to smoked salmon. In addition, not only the aerial parts, but also the roots of this plant have been used in cooking and traditional medicine for a long time [1,2,3]. 

In addition to their nutritional value, capers may also have healing properties. Numerous in vivo and in vitro studies indicate that they may possess various biological and antioxidant properties [1,2,3], which may be due to the chemical content, such as the presence of phenolic compounds. Few papers have described the chemical content and biological properties of capers, especially *C. spinosa* fruits. The present paper reviews the current literature concerning the biological properties of the fruits, leaves, roots, seeds and buds of capers, as well as their toxicity. This literature was obtained by a search of PubMed, Web of Knowledge, ScienceDirect, Sci Finder, Web of Science, CNKI, and Scopus; with extra papers being identified by manually reviewing the references. The search was restricted to English language publications. The last search was run on 30 November 2022. Different terms were used: “caper”, “*C. spinosa*”, “*Capparis spinosa* L.”, “*Capparis spinose*”, “caper’s phenolic compounds”, “*Capparis genus*”, “biological activity of caper”, and “toxicity of caper”.

## 2. Phytochemical Characteristics of *C. spinosa*

Various chemical compounds have been identified in the organs of *C. spinosa*, including alkaloids, phenolic compounds, glucosinolates, sterols, terpenes, terpenoids, fatty acids, saponins, carotenoids and tocopherols. For example, the alkaloids stachydrin, tetrahydroquinoline, and caprisine have been identified in the stems and fruits [3,5,6,7,8,9]. The main phytochemical constituents of the plant and their biological properties are presented in Figure 1.

All *Capparis* species are sources of sulfur-rich metabolites called glucosinolates, which are derived from amino acids. They can be divided into three groups according to their amino acid precursor: indolic, benzonic, and aliphatic. In capers, the most widely distributed is methyl glucosinolate or glucocaprin, constituting 2.05 ± 0.25 mg/g dry weight (dw) [5]. This is accompanied by other glucosinolates, such as glucobrassicin (232 ± 18 µg/g dw) and 4-hydroxyglucobrassicin (89 ± 12 µg/g dw), as well as other compounds present at very low concentrations (0.5–1.5 µg/g dw) [5]. 

A wide range of phenolic compounds, including caffeic acid, catechin, chlorogenic acid, ferulic acid, kaempferol, luteolin, quercetin, resveratrol, rutin, and vanillic acid have been identified in the fruits and leaves of *C. spinosa* [3,5,6,7,8,9]. Interestingly, capers appear to be the richest known natural source of quercetin: the most widely-consumed dietary flavonoid. Indeed, Mohebali et al. [10] have reported high concentrations of quercetin in *C. spinosa* leaves and fruits. It also appears that the pickling process promotes the conversion of rutin to quercetin, and canning may also increase quercetin concentrations. For example, quercetin is present at 520 mg/100 g in canned capers, and 323 mg/100 g in raw capers [10]. 

Wojdyło et al. [11] have studied the phenolic components of *C. spinosa* flower buds at six stages of development: “nonpareilles”, “surfines”, “capucines”, “capotes”, “fines”, and “gruesas” in two cultivars in Spain. Their findings indicate that the total phenolic compound content varied from 3256 to 10,720 mg/100 g dry weight (DW), depending on the stage of growth and genotype. For example, “nonpareilles” had the highest content. Flavonol content also varies according to stage, with “surfines” and “nonpareilles” being richer in these compounds than “fines” and “gruesas” [11].

Sonmezdag et al. [7] have found that fermentation drastically decreases the total amount of aromatic compounds, i.e., from 62,616 to 21,471 µg/mL, in *C. spinosa* L. They also note that this is accompanied by an increase in kaempferol and quercetin concentrations due to the production of enzymes by lactic acid bacteria [7].

Lo Bosco et al. [3] have studied the nutraceutical value of salt-fermented capers (*C. spinosa* L.) collected from five areas of Pantelleria Island (Scauri, Rekhale, Tracino, Barone, and Bugeber). The qualitative and quantitative profiles of the secondary metabolites in hydrophilic extracts were studied by HPLC-electrospray ionization/mass spectrometry. They found that the Bugeber sample had a significantly different level of total phenolic compounds (596.92 ± 53.15 mg gallic acid equivalent (GAE)/100 g DM) compared to those from Barone, Scaurim and Rekhale. They have also reported the presence of 24 other compounds, including various flavonol derivatives and glucoinolates. 

*C. spinosa* also contains vitamin C, carotenoids (lutein and β-carotene), tocopherols (α- and γ-tocopherol), and various fatty acids (palmitic, linolenic, linoleic, and oleic acid). 

Methyl, isopropyl and set-butyl isothiocyanates are the major volatile oils in the fruits and roots of *C. spinose*. Recently, Alipour et al. [12] have demonstrated that essential oils from the fruits of Iranian *C. spinose* possesses 31 of these compounds, including isopropyl isothiocyanate (5.5–13.7%), methyl sulfonyl heptyl isothiocyanate (4.6–156%), butyl isothiocyanate (3.6–10.6%), and other. The phytochemical constituents of *C. spinosa* are present in greater detail in various review papers [6,13,14,15].

## 3. Biological Properties of *C. spinosa*

Capers have been used in traditional medicine since ancient times for their antifungal, diuretic, antioxidant, anti-inflammatory, antidiabetic, antihypertensive, and antihepatoxic actions. These biological activities have been ascribed to various bioactive compounds, including certain phenolic compounds, lipids, indoles and alkaloids [3,5,6,7,8,9].

### 3.1. Anti-Inflammatory Properties

A number of studies have highlighted the anti-inflammatory potential of *C. spinasa* organs, including its leaves and fruits. For example, Moutia et al. [16] have demonstrated that crude *C. spinosa* leaf extract (0.0824 ± 0.0012 mg gallic acid equivalent phenolic compounds per gram) and the aqueous fraction stimulate an anti-inflammatory response in human peripheral blood mononuclear cells from healthy donors via the inhibition of proinflammatory IL-17. The findings also suggest that treatment induces IL-17 gene expression in vitro. Hamuti et al. [17] have also found that the ethanol extract obtained from *C. spinasa* fruits has anti-inflammatory properties.

Maresca et al. [18] have reported that the hydroalcoholic extract (300 mg/kg) obtained from *C. spinosa* roots has anti-inflammatory properties in vivo. El Azhary et al. [19] have also indicated that the methanol extract and hexane fraction from caper leaves containing compounds with anti-inflammatory properties can orientate the immune response mediated by CD4+ T cells in vivo, i.e., by a contact hypersensitivity model in Swiss albino mice. The tested plant preparations were administered by injection for 2, 3, 4, and 7 days at different doses, e.g., 1.07 and 0.428 g/kg body weight for methanol extract. 

Recently, Rahimi et al. [20] studied the effects of the hydroethanolic extract obtained from aerial parts of *C. spinosa* on lipopolysaccharide-induced inflammation in vivo and in vitro. The extract is known to contain various flavonoids, mainly quercetin and kaempferol derivatives. In the in vivo model, rats (n = 40) were supplemented with *C. spinosa* extract (100 and 300 mg/kg/day for four weeks); the in vitro studies used different concentrations of plant extract from 10 to 300 µg/mL. The expression of anti-inflammatory and pro-inflammatory cytokines was measured by ELISA and Real-Time PCR. The results indicate that the *C. spinosa* extract reduced the levels of pro-inflammatory cytokines in both models. 

Zhu et al. [21] have reported that treatment with the aqueous *C. spinosa* fruit extract (200 and 400 mg/kg/day, for seven days) has anti-inflammatory effects in mice (n = 7) with ulcerative colitis, a chronic inflammatory disease. The tested extract reduced the expression of proinflammatory cytokines (TNF-α, IL-6, and IL-1β) and was found to have antioxidant properties. 

Other studies have found that hydroalcoholic extracts from *C. spinosa* fruits and leaves appear to have the potential to downregulate genes controlling inflammation in Alzheimer’s disease, as demonstrated in rats injected with amyloid-beta peptide (n = 12) [10]. Yosri et al. [22] have reported that the administration of ethanolic extract of *C. spinosa* flowers decreases inflammation in a rat arthritis model by moderating the concentrations of inflammatory mediators, osteoclasts and chondrocytes. 

### 3.2. Antihyperglycemic, Antioxidant and Hypolipidemic Activities in Diabetic and Non-Diabetic Models

Several papers have found that aqueous and hydroalcoholic extracts from the fruits, leaves, seeds and roots of *C. spinosa* possess antihyperglycemic properties at a wide range of doses (20–800 mg/kg for 12 to 60 days). For example, the four-week administration of 0.2 and 0.4 g/kg/day root extract demonstrated anti-diabetic properties in diabetic rats [23]. Such antihyperglycemic properties have been observed in both animal models and clinical trials. For example, the hydroalcoholic extract of *C. spinosa* fruits was shown to have antihyperglycemic action in type 2 diabetes patients (*n* = 28) receiving 400 mg extract three times a day for two months [24]. 

Vahid et al. [6] propose that the antihyperglycemic mechanism of *C. spinosa* root and leaf extracts is based on reducing carbohydrate absorption from the small intestine; in addition, the fruit extract may inhibit gluconeogenesis in the liver and enhance glucose uptake by the tissues. It was also found that the three-week consumption of *C. spinosa* may reduce damage to beta cells. 

However, the effect of the extracts on insulin release from beta cells is controversial [1]. A recent study by Assadi et al. [25] looked at the antidiabetic potential of hydro-ethanolic extracts of *C. spinosa* fruits in rats with type 2 diabetes. They observed that the fruit extract decreased glucose intolerance. 

Rakhshandeh et al. [26] studied the effect of *C. spinosa* extract on ischemic stroke caused by the middle cerebral artery occlusion in Wistar rats. The rats received this extract orally, once a day for seven days, before induction by middle cerebral artery occlusion. The *C. spinosa* extract was found to effectively protect the middle cerebral artery occlusion injury via attenuation or the reduction of oxidative stress. The tested extract reduced lipid peroxidation, expressed by malondialdehyde assay (MDA), and increased the level of thiols in the brain tissues compared to the control group. 

Other studies suggest that the leaves, buds and seeds also have antioxidant potential in other diseases, including Alzheimer’s disease [27,28,29]. In addition, ethanolic *C. spinosa* leaf extract (50–500 mg/kg bw) was found to reduce oxidative stress induced by potassium bromate (KBrO_3_; 150 mg/kg bw) in mice; in this case, the level of oxidative stress was measured by various biomarkers, including the activity of catalase, superoxide dismutase, glutathione peroxidase and lipid peroxidase, as well as reactive oxygen species (ROS) concentration. In addition, the tested extract also demonstrated a protective action against genotoxicity stimulated by KBrO_3_ [30].

Interestingly, *C. spinosa* fruit extract (300 mg/kg) was found to bestow similar protective effects as vitamin E (200 mg/kg) and quercetin (10 mg/kg) on kidney, liver and heart function by decreasing oxidative stress. In this experiment, the plant extract and chemical compounds were orally administered for two weeks. This study included rats treated with monosodium glutamate [31].

Weight loss was notably observed in diabetic rats fed with *C. spinosa* fruit extract (20 mg/kg) after two weeks [32]. However, these findings need to be confirmed in further studies on diabetic and non-diabetic patients. More details about the antidiabetic properties of *C. spinosa* and its components are given in a review by Vahid et al. [6]. For example, the authors indicate that the antihyperglycemic effects of *C. spinosa* include reducing carbohydrate absorption from the small intestine, inhibiting gluconeogenesis in the liver, enhancing glucose uptake by tissues, and beta cell protection/regeneration. In addition, this plant may ameliorate cardiovascular disorders, nephropathy and liver damage in different models of diabetes.

The biological properties and the anti-diabetic potential of various caper organs (fruits, leaves, roots, seeds, and buds) are summarized in Table 1. However, although anti-diabetic activity has been observed in animal models, only one in vivo study has been performed in type 2 diabetic human patients. In addition, while *C. spinosa* extract has been found to have beneficial effects on lipid profiles in various models based on diabetic animals, no study has examined the mechanisms underlying the hypolipidemic properties of *C. spinosa* (Table 1). However, Huseini et al. [24] reported a decrease in triglyceride concentration in type 2 diabetic patients supplemented with 400 mg fruit extract three times a day for 60 days. The authors suggest that the tested extract decreases the activity of 3-hydroxy-3-methyl-glutaryl coenzyme A reductase, an important enzyme in the synthesis of cholesterol. In addition, Sardari et al. [33] have observed that daily consumption of caper fruit pickle (40–50 g for seven days) has synergistic actions with atorvastatin in hyperlipidemic human patients (*n* = 60). 

In contrast, Vahid et al. [6] have that found *C. spinosa* oxymel (10 mL three times a day for three months) did not enhance the effects of hypolipidemic or hypoglycemic drugs in diabetic patients with metabolic syndrome. *C. spinosa* oxymel was prepared by adding the hydroalcoholic extract of *C. spinosa* fruits to a mixture of grape vinegar and lactulose.

### 3.3. Anti-Hypertensive Action

Ali et al. [51] found that aqueous *C. spinosa* fruit extract (150 mg/kg, for 20 days) decreases systolic blood pressure in hypertensive rats, suggesting that it may have anti-hypertensive potential. The authors attribute the anti-hypertensive properties to the increased excretion of renal electrolytes and the inhibition of plasma angiotensin-converting enzyme (ACE) activity. Ali et al. [51] also found that *C. decidua* stem has anti-hypertensive activity in vitro and in vivo (arsenic-induced hypertension in rats). The anti-hypertensive action was found to be associated with the endothelium-dependent and Ca^2+^ antagonistic pathways.

### 3.4. Anti-Hepatotoxic Action

El-Hawary et al. [46] found methanolic *C. spinosa* leaf extract (100–4000 mg/kg/day) to demonstrate anti-hepatotoxic properties in mice experimentally infected with *Schistosoma monsoni,* and it was harmless for the organism. Kalantari et al. [47] also found hydroalcoholic *C. spinosa* leaf extract (400 mg/kg/day, for five days) to have hepatoprotective properties against tert-butyl hydroperoxide-induced acute liver damage in Swiss albino mice (*n* = 56). The authors attribute this to the free radical-scavenging properties of the extract, which contains quercetin and other phenolic compounds. Tlili et al. [52] report that methanolic *C. spinosa* leaf extract demonstrated hepatoprotective, antioxidant and nephroprotective properties in albino rats. Gadgoli and Mishra [53] also found that aqueous *C. spinosa* aerial extract has anti-hepatotoxic effects in vitro and in vivo.

Khavari et al. [54] reported that the daily consumption of pickled caper fruit (40–50 g/day, for 12 weeks) has a beneficial action in non-alcoholic fatty liver disease patients (*n* = 44). Akbari et al. [42] report *C. spinosa* fruit extract (20 mg/kg/day, for 12 weeks) to have beneficial effects against non-alcoholic steatohepatitis in Wistar rats receiving a high-fat diet, and attribute this to the upregulation of fibroblast growth factor 21 (FGF21).

### 3.5. Anti-Cancer Activity

A study of the anti-cancer potential of hydroalcoholic extract (quercetin was the main phenolic compound) from aerial parts of *C. spinosa* found it to significantly decrease the viability of breast cancer (MCF7), HeLa and osteosarcoma (Saos) cell lines [50], as determined by MTT assay. The most effective dose against cancer cells was 250 µg/mL for 72 h. The tested extract also demonstrated high antioxidant potential, measured by a ferric antioxidant power assay (FRAP). The study used five various concentrations of extract (62.5, 125, 250, 500 and 1000 µg/mL) and three incubation times (24, 48 and 72 h).

Ji and Yu [43] have observed that the n-butanol extract of *C. spinosa* fruits (1–100 µg/mL) not only reduces cell viability in a dose-dependent manner, but also induces apoptosis in the SGC-7901 human gastric carcinoma cell line. They propose that this activity occurs through a mitochondrial pathway involving mPTP open, caspase-3 and -9 activation, and cytochrome C release. In another study, Ji and Yu [44] indicate that the extract of *C. spinosa* fruits also induces SGC-7901 cell apoptosis by upregulating B-cell lymphoma (BCL-2)-associated X protein expression, and downregulating that of BCL-2. Furthermore, Yi and Yu [55] report that seleno- *C. spinosa* polysaccharide obtained by selenylation significantly increases cytotoxicity to tumor cells, including SGC-7901 cells.

It is an important to note that all cited studies are in vitro only (cell lines cytotoxicity) and therefore have serious limitations, as the relevance to in vivo anticancer or even carcinogenic preventive properties are unverified. The literature is full of data on cell lines using many common foodstuffs and trivial phytochemicals, and still most of them are not developed into an anticancer drug.

### 3.6. Other Biological Actions

Some papers indicate that various parts of *C. spinosa*, including the leaves, roots, fruits, stem barks and shoots, have anti-bacterial properties [56,57,58,59]. In addition, Lam and Ng [60] propose that *C. spinosa* may have an antifungal activity and exhibit inhibitory effects on human immunodeficiency virus (HIV-1) reverse transcriptase. In addition, it has been found that the ubiquitous flavonoid quercetin in 1% caper extract activates atypical KCNQ potassium channels (K_v_) [9].

Rakhshandeh et al. [41] have reported that the hexane, ethyl acetate, and water fractions of *C. spinosa* hydro-alcoholic extract appear to have hypnotic properties when administered at three concentrations (30, 60, and 120 mg/kg body weight). Diazepam (3 mg/kg body weight) was used as a positive control, and saline as a negative control.

## 4. Toxicity of *C. spinosa*

The data regarding the toxicity of *C. spinosa* is controversial. Some authors indicate that various preparations from *C. spinosa* may sometimes induce side effects. For example, Heidari et al. [61] have found *C. spinosa* fruits to be nephrotoxic in rats. However, Kazemian et al. [23] indicate that the consumption of capers is safe and does not elicit toxic effects on the liver, and Huseini et al. [24] do not note any hepatotoxicity or nephrotoxicity of *C. spinosa* fruit extract (400 mg three times a day, for two months) among diabetic patients. Furthermore, Moutia et al. [16] note that *C. spinosa* leaf preparations do not induce cell toxicity, even when applied at concentrations of 700 µg/mL. However, well-designed clinical trials are necessary. Rakhshandeh et al. [41] have also reported that the *C. spinosa* extract did not appear to induce any cytotoxic action in L929 fibroblast cells using the MTT test.

## 5. Conclusions

Nevertheless, biological activities have been observed, especially in animal-based models (Table 1). Only a single study has examined the hypolipidemic properties of pickled caper fruit consumption in hyperlipidemic human patients [33]. 

In addition, the biological mechanisms by which caper consumption influences human health remains unclear and poorly defined in the scientific literature. Despite this, the compounds in *C. spinosa*, particularly its phenolic compounds, may play an important role in these mechanisms. Therefore, it is still too early to offer more than a tentative assessment of the beneficial role of *C. spinosa* and its preparations on the treatment or prophylaxis of diseases such as cancer, hypertension, and diabetes. As such, further randomized clinical trials are needed.

## Figures and Tables

**Figure 1 nutrients-15-00623-f001:**
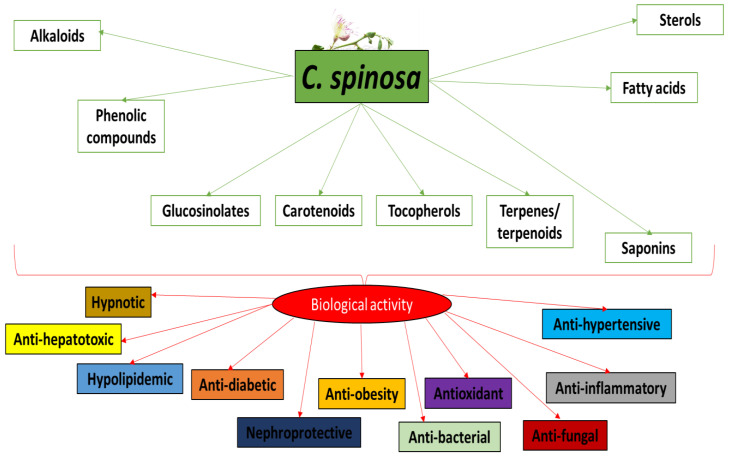
The major chemical compounds identified in *C. spinosa* and their biological properties.

**Table 1 nutrients-15-00623-t001:** Biological properties of different parts (fruits, leaves, roots, seeds, and buds) of *C. spinosa* in in vitro and in vivo models.

Part of *C. spinosa*	Type of Extract/Fraction or Other/Dose	Experimental Model	Biological Properties	References
Fruits	Hydroalcoholic extract; 300 mg/kg; 12 days	Diabetic rats (in vivo model)	Antidiabetic properties	[34]
Fruits	Hydroalcoholic extract; 200 and 800 mg/kg; 28 days	Diabetic rats (in vivo model)	Antidiabetic activity and hypolipidemic properties	[35]
Fruits	Hydroalcoholic extract; 200 and 800 mg/kg; 28 days	Diabetic rats (in vivo model)	Antidiabetic properties	[36]
Fruits	Hydroalcoholic extract; 20 and 30 mg/kg; 28 days	Diabetic rats (in vivo model)	Antidiabetic activity and hypolipidemic properties	[37,38]
Fruits	Alcoholic extract; 20 mg/kg; 14 days	Diabetic rats (in vivo model)	Antidiabetic activity and anti-obesity properties	[32]
Fruits	Alcoholic extract; 20 mg/kg; 28 days	Diabetic rats (in vivo model)	Antidiabetic activity and hypolipidemic properties	[39]
Fruits	Hydroalcoholic extract; 400 mg/kg three times a day; 60 days	Type 2 diabetic patients (in vivo model)	Antidiabetic activity and hypolipidemic properties	[24]
Fruits	Ethanolic extract; 10, 50, and 100 g/mL	Systemic sclerosis dermal fibroblasts (in vitro model)	Antioxidant properties	[27]
Fruits	Hydro-ethanolic extract; 400 mg/kg	Diabetic rats (in vivo model)	Antidiabetic activity and antioxidant properties	[25]
Fruits	Aqueous extract; 150 mg/kg; 20 days	Hypertensive rats (in vivo model)	Anti-hypertensive properties	[40]
Fruits	Hydro-alcoholic extract; 30, 60, and 120 mg/kg	Mice (in vivo model)	Hypnotic activity	[41]
Fruits	Daily caper fruit pickle consumption (40–50 g for eight)	Hyperlipidemic human patients (in vivo model)	Hypolipidemic properties	[33]
Fruits	Extract; 20 mg/kg/day; for 12 weeks	Wistar rats with non-alcoholic steatohepatitis (in vivo model)	Anti-hepatotoxic properties	[42]
Fruits	Ethanol extract; 0.17–1.7 mg/mL	Dendritic cells (in vitro model)	Anti-inflammatory properties	[17]
Fruits	Water extract; 200 and 400 mg/kg/day; 7 days	Ulcerative colitis mice (in vivo model)	Anti-inflammatory activity and antioxidant properties	[21]
Fruits	Hydroalcoholic extract; 6 weeks	Rat model of Alzheimer disease (in vivo model)	Anti-inflammatory properties	[10]
Fruits	N-butanol extract; 1–100 µg/mL	SGC-7901 cells (in vitro model)	Anti-cancer properties	[43,44]
Leaves	Phenolic extract; 15 and 25 mg/kg; 28 days	Diabetic rats (in vivo model)	Antidiabetic properties	[45]
Leaves	Methanol extract; 1.07 and 0.428 g/kg, for 2,3,4 and 7 days	Swiss albino mice (in vivo model)	Anti-inflammatory properties	[19]
Leaves	Aqueous fraction and crude extract; 10, 100, and 500 µg/mL	Human peripheral blood mononuclear cells (in vitro model)	Anti-inflammatory properties	[16]
Leaves	Methanol extract; 100–4000 mg/kg/day	Mice infected with S. monsoni (in vivo model)	Anti-hepatotoxic properties	[46]
Leaves	Hydroalcoholic extract; 400 mg/kg/day; 5 days	Swiss albino mice (in vivo model)	Anti-hepatotoxic activity and antioxidant properties	[47]
Leaves	Ethanolic extract; 50–500 mg/kg bw	Mice treated with KBrO_3_ (in vivo model)	Antioxidant properties	[30]
Roots	Hydroalcoholic extract; 200 and 400 mg/kg; 28 days	Diabetic rats (in vivo model)	Antidiabetic activity and hypolipidemic properties	[23]
Roots	Extract; 300 mg/kg	Male Spraque-Dawley rats (in vivo model)	Anti-inflammatory properties	[18]
Roots	Hydroalcoholic extract; 0.2 and 04 g/kg/day; 4 weeks	Diabetic rats (in vivo model)	Antidiabetic properties	[23]
Seeds	Hydroalcoholic extract; 200, 400 and 800 mg/kg; 21 days	Diabetic rats (in vivo model)	Antidiabetic activity and hypolipidemic properties	[48]
Seeds	Phenolic extract; 50, 100, and 200 mg/kg; 8 weeks	Mice with Alzheimer’s disease (in vivo model)	Antioxidant properties	[49]
Buds	Aqueous extract; 10 and 30 mg/kg; 7 days	Rats with Alzheimer disease (in vivo model)	Antioxidant properties	[29]
Buds	Phenolic extract	α-glucosidase and α-amylase (in vitro model)	Antidiabetic properties	[11]
Flowers	Ethanolic extract; 7, 14, and 28 mg/kg/day; 15 days	Arthritis rat model (in vivo model)	Anti-inflammatory properties	[22]
Aerial parts of plant	Hydroalcoholic extract; 62.5–1000 µg/mL; 24, 48 and 72 h	MCF7, HeLa and Saos cancer lines (in vitro model)	Anti-cancer activity and antioxidant properties	[50]
Aerial parts of plant	Hydroalcoholic extract; 100 and 300 mg/kg/day; 4 weeks	Rats (in vivo model)	Anti-inflammatory properties	[20]
Aerial parts of plant	Hydroalcoholic extract; 10–300 µg/mL	Rat cells (in vitro model)	Anti-inflammatory properties	[20]

## Data Availability

Not applicable.

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
