# Peer review of "The Current State of Knowledge about the Biological Activity of Different Parts of Capers"

_nutrients, 2023, doi:10.3390/nu15030623_

Round 1
Reviewer 1 Report
This article reviews the biological properties of different parts of the plant C. spinosa.
There are several aspects of this paper that needs to be précised / corrected:
Abstract
Line 12-13 It will be better just to left “ do not contain a large number of calories “, because if you put exact number it will need reference not to put exact number, since such statement needs reference.
Line 17-18 This sentence is not necessary since the previous one stated the purpose of the review
Introduction
Line 22-26 all the statements in this paragraph need references
Line 28-29 “although mainly” is better to be replaced with” especially”
Line 31-32 “watermelon in China” is actually wild watermelon(Ao, M.; Gao, Y.; Yu, L. Advances in studies on constituents and their pharmacological activities of Capparis spinosa. Chin. Tradit. Herb. Drug 2007, 38, 463–467.)
Besides, there are no references for the other local names.
Line 33-39 the statements in this paragraph need references
Line 57 “ fruits, leaves, roots, seeds and buds of capers” but in the abstract is not mentioned buds and seeds
2. Phytochemical characteristics of C. spinosa
Line 72-73 need references
Line 82-83 need references
Line 85-87 need references
Line 100-106 there are too many details on the test performed by Lo Bosco et al. (2019). It will be better if the paragraph is re-written, focusing on the most important findings of the cited paper
3.1. Anti-inflammatory properties
Line 144-152 the paragraph contains too much additional information not related to the general findings, and at the same time it is not clear whether the extract have pro- or anti- inflammatory effect or reduces all the types of cytokines
3.2. Antihyperglycemic, antioxidant and hypolipidemic activities in diabetic and non-diabetic models
Line 167 “to have” will be better replaced by possesses in order to avoid repetition
Line 177 maybe you mean “ by the tissues”?
Line 179-184 This paragraph is unclear. It needs to be re-written
Line 188 followed
Table 1 The following references are in the table but not in the reference list at the end of the paper :
Hashemnia et al., 2012 ; Rahmani et al., 2013 ; Taghavi et al., 2014 ; Negahdarizadeh et al., 2011; Mohammadi et al., 2012 ; Eddouks et al., 2004 ; Jalali et al., 2016 ; Ali et al., 2007 ; Feng et al., 2011 ; Ji and Yu, 2014; Ji and Yu 2015 ; Oudah et al., 2014; Kalantari et al., 2017 but in references is 2018; Kazamian et al., 2015 ; Nazari et al., 2014 ; Turgut et al., 2015 ; Panico et al., 2005 ; Rahimi et al., 2020
3.4. Anti-hepatotoxic action
Line 246 “and to be safe” - maybe you mean it was harmless for the organism?
3.6. Other biological actions
Line 288-290 is better suited in 4. Toxicity of C. spinosa
5. Conclusion
Line 302-303 This statement is not correct because there are actually a lot of papers regarding the photochemistry of capers.
Reference â„– 46 it is Zhu not Zu
In General: References in the text are not prepared according to Nutrients guidelines.
Among the other things that needs to be improved I will recommend the authors to check again in the scientific database using searching terms like “capers phytochemicals” . Adding more data will give more value to this review.
Author Response
Rev. 1
This article reviews the biological properties of different parts of the plant C. spinosa.
Thank you for reviewing the manuscript and providing such helpful comments. All of them have been taken into consideration when revising the manuscript.
There are several aspects of this paper that needs to be précised / corrected:
Abstract
Line 12-13 It will be better just to left “ do not contain a large number of calories “, because if you put exact number it will need reference not to put exact number, since such statement needs reference.
Response: I have removed this information.
Line 17-18 This sentence is not necessary since the previous one stated the purpose of the review
Response: I have removed this sentence.
Introduction
Line 22-26 all the statements in this paragraph need references
Response: I have added references: “Nabavi et al., 2016; Zhang and Ma, 2018; Lo Bosco et al., 2019”.
Line 28-29 “although mainly” is better to be replaced with” especially”
Response: I have corrected. Now, it is: “especially”.
Line 31-32 “watermelon in China” is actually wild watermelon(Ao, M.; Gao, Y.; Yu, L. Advances in studies on constituents and their pharmacological activities of Capparis spinosa. Chin. Tradit. Herb. Drug 2007, 38, 463–467.)
Besides, there are no references for the other local names.
Response: I have added references: “Nabavi et al., 2016; Zhang and Ma, 2018; Lo Bosco et al., 2019”, and Ao et al., 2007”.
Line 33-39 the statements in this paragraph need references
Response: I have added references: “Nabavi et al., 2016; Zhang and Ma, 2018; Lo Bosco et al., 2019”.
Line 57 “ fruits, leaves, roots, seeds and buds of capers” but in the abstract is not mentioned buds and seeds
Response: I have added “buds, seeds” in Abstract.
- Phytochemical characteristics of C. spinosa
Line 72-73 need references
Response: I have added: “(Mohebali et al., 2016)”.
Line 82-83 need references
Response: I have added: “(WojdyÅ‚o et al., 2019)”.
Line 85-87 need references
Response: I have added: “(Sonmezdag et al. 2019)”.
Line 100-106 there are too many details on the test performed by Lo Bosco et al. (2019). It will be better if the paragraph is re-written, focusing on the most important findings of the cited paper
Response: I have changed this paragraph.
3.1. Anti-inflammatory properties
Line 144-152 the paragraph contains too much additional information not related to the general findings, and at the same time it is not clear whether the extract have pro- or anti- inflammatory effect or reduces all the types of cytokines
Response: I have added more information about it: “Recently, Rahimi et al. (2020) studied the effects of the hydroethanolic extract obtained from aerial parts of C. spinosa on lipopolysaccharide-induced inflammation in vivo and in vitro. The extract is known to contain various flavonoids, mainly quercetin and kaempferol derivatives. In the in vivo model, rats (n=40) were supplemented with C. spinosa extract (100 and 300 mg/kg/day for four weeks); the in vitro studies used different concentrations of plant extract from 10 to 300 µg/mL. The expression of anti-inflammatory and pro-inflammatory cytokines was measured by ELISA and Real-Time PCR. The results indicate that the C. spinosa extract reduced the levels of pro-inflammatory cytokines in both models.”
3.2. Antihyperglycemic, antioxidant and hypolipidemic activities in diabetic and non-diabetic models
Line 167 “to have” will be better replaced by possesses in order to avoid repetition
Response: I have changed. Now, it is: “Several papers have found that aqueous and hydroalcoholic extracts from the fruits, leaves, seeds and roots of C. spinosa possess antihyperglycemic properties at a wide range of doses (20 mg/kg – 800 mg/kg for 12 to 60 days).”
Line 177 maybe you mean “ by the tissues”?
Response: I have corrected. Now, it is: “in addition, the fruit extract may inhibit gluconeogenesis in the liver and enhance glucose uptake by the tissues.”
Line 179-184 This paragraph is unclear. It needs to be re-written
Response: I have changed this paragraph.
Line 188 followed
Response: I have changed this paragraph.
Table 1 The following references are in the table but not in the reference list at the end of the paper :
Hashemnia et al., 2012 ; Rahmani et al., 2013 ; Taghavi et al., 2014 ; Negahdarizadeh et al., 2011; Mohammadi et al., 2012 ; Eddouks et al., 2004 ; Jalali et al., 2016 ; Ali et al., 2007 ; Feng et al., 2011 ; Ji and Yu, 2014; Ji and Yu 2015 ; Oudah et al., 2014; Kalantari et al., 2017 but in references is 2018; Kazamian et al., 2015 ; Nazari et al., 2014 ; Turgut et al., 2015 ; Panico et al., 2005 ; Rahimi et al., 2020
Response: I have corrected.
3.4. Anti-hepatotoxic action
Line 246 “and to be safe” - maybe you mean it was harmless for the organism?
Response: I have changed. Now, it is: “El-Hawary et al. (2018) found methanolic C. spinosa leaf extract (100 – 4000 mg/kg/day) to demonstrate anti-hepatotoxic properties in mice experimentally infected with Schistosom monsoni and it was harmless for the organism.”
3.6. Other biological actions
Line 288-290 is better suited in 4. Toxicity of C. spinosa
Response: I have added this information – the chapter “Toxicity of C. spinose”.
- Conclusion
Line 302-303 This statement is not correct because there are actually a lot of papers regarding the photochemistry of capers.
Response: I have deleted this sentence.
Reference â„– 46 it is Zhu not Zu
Response: I have corrected. Now, it is “Zhu”.
In General: References in the text are not prepared according to Nutrients guidelines.
Response: I have corrected.
Among the other things that needs to be improved I will recommend the authors to check again in the scientific database using searching terms like “capers phytochemicals” . Adding more data will give more value to this review.
Response: I have added more information about it. For example, “Recently, results of Alipour et al. (2021) have demonstrated that essential oil from fruits of Iranian C. spinose possesses 31 these compounds, including isopropyl isothiocyanate (5.5-13.7%), methyl sulfonyl heptyl isothiocyanate (4.6-156%), butyl isothiocyanate (3.6-10.6%), and other. The phytochemical constituents of C. spinosa are presents in more detail in a review by Vahid et al. (2017).”
Reviewer 2 Report
The present review paper deals with the current state of knowledge about the biological activity of different parts of capers. The scope of the study is well described. The text is easy to follow while appropriate details are given in order someone, not familiar with the subject, to fully understand. The subject is well discussed, including the current literature and appropriate references.
I would like to indicate two points:
- - Pg1, lns 33-39: Appropriate reference(s) should be added.
- - Minor revision in English is needed.
Author Response
Rev. 2
The present review paper deals with the current state of knowledge about the biological activity of different parts of capers. The scope of the study is well described. The text is easy to follow while appropriate details are given in order someone, not familiar with the subject, to fully understand. The subject is well discussed, including the current literature and appropriate references.
Thank you for reviewing the manuscript and providing such helpful comments. All of them have been taken into consideration when revising the manuscript.
I would like to indicate two points:
- - Pg1, lns 33-39: Appropriate reference(s) should be added.
Response: I have added references: “Nabavi et al., 2016; Zhang and Ma, 2018; Lo Bosco et al., 2019”.
- - Minor revision in English is needed.
Response: The native speaker corrected this manuscript.
Reviewer 3 Report
A clearly written article presenting numerous aspects of the action of compounds derived from capers, although due to the extensiveness of the subject, it cannot discuss, for example, the importance of organic isothiocyanates, the presence of which has been signaled.
Author Response

(The authors gave the same response as above.)

Round 2
Reviewer 1 Report
Line 26 - Please, specify in which part of Bible it has been mentioned.
Line 96-106 Please, remove the unnecessary description of the methods for antioxidant activity evaluation. After all, the fact that there is antioxidant activity is enough especially when there is also phytochemistry included.
Line 111 “31 these” add “of”
Line 171-176 This paragraph is a bit confusing. It starts with anti-diabetic effect, but it actually focus on antioxidant activity and just at the end it mentioned decrease in glucose intolerance. It needs to be focused more on the anti-diabetic effect.
Line 277-280 It is better suited in “Other biological actions”, not in the toxic chapter. Previously, I proposed to move only the part with cytotoxicity here.
Reference 17 the name of the first author is Rahimi
Author Response
Thank you for reviewing the manuscript and providing such helpful comments. All of them have been taken into consideration when revising the manuscript.
Line 26 - Please, specify in which part of Bible it has been mentioned.
Response: I have added this information: “(Book of Ecclesiastes)”
Line 96-106 Please, remove the unnecessary description of the methods for antioxidant activity evaluation. After all, the fact that there is antioxidant activity is enough especially when there is also phytochemistry included.
Response: I have removed this information.
Line 111 “31 these” add “of”
Response: I have corrected. Now, it is:”31 these of compounds”.
Line 171-176 This paragraph is a bit confusing. It starts with anti-diabetic effect, but it actually focus on antioxidant activity and just at the end it mentioned decrease in glucose intolerance. It needs to be focused more on the anti-diabetic effect.
Response: I have corrected this paragraph. Now, it is: “However, the effect of the extracts on insulin release from beta cells is controversial (1). A recent study by Assadi et al. (22) studied the antidiabetic potential of hydro-ethanolic extracts of C. spinosa fruits in type 2 diabetic rats. They observed that the fruit extract decreased glucose intolerance.”
Line 277-280 It is better suited in “Other biological actions”, not in the toxic chapter. Previously, I proposed to move only the part with cytotoxicity here.
Response: I have corrected. Now, it is:”31 these of compounds”.
Reference 17 the name of the first author is Rahimi
Response: I have corrected. Now, it is:”Rahimi”.